# Long-Term Evaluation of Gastroesophageal Reflux in Neonates with and without Preventive Anti-reflux Surgery at the Time of Congenital Diaphragmatic Hernia Repair

**DOI:** 10.3390/children9081137

**Published:** 2022-07-29

**Authors:** Christoph von Schrottenberg, Susanne Deeg, Christel Weiss, Rüdiger Adam, Lucas M. Wessel, Michael Boettcher, Katrin B. Zahn

**Affiliations:** 1Department of Pediatric Surgery, University Hospital Mannheim, University of Heidelberg, 68167 Mannheim, Germany; susanne.deeg@umm.de (S.D.); lucas.wessel@medma.uni-heidelberg.de (L.M.W.); michael.boettcher@umm.de (M.B.); katrin.zahn@umm.de (K.B.Z.); 2Department of Medical Statistics and Biomathematics, Medical Faculty Mannheim, University of Heidelberg, 68167 Mannheim, Germany; christel.weiss@medma.uni-heidelberg.de; 3Department of Pediatrics, University Hospital Mannheim, University of Heidelberg, 68167 Mannheim, Germany; ruediger.adam@umm.de; 4ERNICA-Center, 68167 Mannheim, Germany

**Keywords:** gastroesophageal reflux, GER, congenital diaphragmatic hernia, CDH, long-term follow-up, fundoplication, preventive anti-reflux surgery

## Abstract

One potential comorbidity after congenital diaphragmatic hernia (CDH) is gastroesophageal reflux (GER), which can have a substantial effect on patients’ quality of life, thriving, and complications later in life. Efforts have been made to reduce gastroesophageal reflux with a preventive anti-reflux procedure at the time of CDH repair. In this follow-up study of neonates participating in a primary RCT study on preventive anti-reflux surgery, symptoms of GER were assessed longitudinally. Long-term data with a median follow-up time of ten years was available in 66 patients. Thirty-one neonates received an initial fundoplication. Secondary anti-reflux surgery was necessary in 18% and only in patients with large defects. It was required significantly more often in patients with intrathoracic herniation of liver (*p* = 0.015) and stomach (*p* = 0.019) and patch repair (*p* = 0.03). Liver herniation was the only independent risk factor identified in multivariate regression analysis. Primary fundopexy and hemifundoplication did not reveal a protective effect regarding the occurrence of GER symptoms, the need for secondary antireflux surgery or the gain of body weight regardless of defect size neither in the short nor in the long term. Symptoms of GER must be assessed carefully especially in children with large defects, as these are prone to require secondary anti-reflux surgery in the long-term. Routine evaluation of GER including endoscopy and impedance measurement should be recommended especially for high-risk patients.

## 1. Introduction

Congenital diaphragmatic hernia (CDH) is a rare and potentially life-threatening defect of the diaphragm. Because of steady advances in the fields of neonatology and pediatric surgery, these patients today have a relatively good chance to survive, with survival rates of 60–81% reported from some high-volume centers [1,2]. Survival is particularly dependent on defect size and accompanying congenital cardiac anomalies [3]. As standardized perinatal management of patients with CDH, including stabilization under ECMO (extracorporeal membrane oxygenation) therapy, has led to improvement of survival rates especially of severe cases of CDH, an increase in the rate of comorbidities can be expected over the next few years [4,5,6,7]. As survival rates have improved, long-term follow-up of these patients has gained particular importance over the last decade. Structured follow-up programs have been implemented to monitor and treat the numerous comorbidities at some centers [8]. These include cardiopulmonary compromise, musculoskeletal deformities such as scoliosis or funnel chest, neurological impairment and gastrointestinal symptoms including nutritional challenges.

Gastroesophageal reflux (GER) is one complication in patients with CDH that can have a severe impact on the quality of life of these children. It can cause recurrent pulmonary infections and obstructive pulmonary function disorders in a patient with already compromised lung function due to lung hypoplasia inherent to CDH. Furthermore, vomiting, feeding problems and thus failure to thrive may result especially in younger children [9,10]. Failure to thrive is known to have a negative impact on neurocognitive development and thus contributes to neurodevelopmental delay in CDH patients [11]. Gastroesophageal reflux can also be the underlying cause for brief, resolved, unexplained events (BRUE) in infancy. Even silent reflux can cause long-term morbidity, as it can lead to dysplasia of the mucosal tissue of the esophagus resulting in Barrett’s esophagus and eventually in an adenocarcinoma of the esophagus, as was described by Steven et al. [12]. Prevalence of gastroesophageal reflux in CDH patients varies strongly in the literature from 52.7% in infants to 35.1% in children older than one year [13,14]. Because of this, many studies have been conducted to identify risk factors in CDH patients that would predict the need for anti-reflux surgery. One study showed that 61% of patients with patch repair of the diaphragmatic defect and 73% of patients with intrathoracic liver herniation developed GER symptoms and required anti-reflux surgery during follow-up in 32% and 38% of cases, respectively. Intrathoracic liver herniation was identified as the only independent risk factor for gastroesophageal reflux and the need for anti-reflux surgery [14]. Another study identified liver herniation and use of patch for diaphragmatic repair each as significant predictors for the need of anti-reflux surgery during follow-up [15]. Furthermore, Cordier et al. showed that antenatal stomach position was the only predictive factor for gastrointestinal morbidity (oral aversion, not on full oral nutrition) at two years of age [16]. Nowadays, most high-risk patients (large defect-size, intrathoracic liver and stomach herniation) are already identified prenatally and should thus be referred to specialized centers for postnatal treatment and structured follow-up. Further studies have looked at the benefit of a simultaneous preventive anti-reflux procedure at the time of surgical repair of the diaphragmatic defect. Chamond et al. performed a simultaneous preventive anti-reflux surgery in 17 neonates simultaneously with CDH repair. They evaluated GER symptoms and the need for secondary anti-reflux surgery longitudinally over a follow-up time of three years and concluded that patients with intrathoracic liver herniation or need for patch repair of the diaphragm profited from the preventive procedure. Patients with preventive anti-reflux surgery suffered from significantly less GER symptoms at one year of follow-up, while 30% of patients without preventive anti-reflux surgery required a fundoplication before the age of six months [17]. At our center, a monocentric, single-blinded randomized controlled study was conducted between 2003 and 2009. Thirty-six patients received a preventive hemifundoplication at the time of CDH repair and were compared to forty-three patients who only received surgical closure of the diaphragmatic defect. Endpoints were gain of weight and gastroesophageal reflux symptoms. An almost significant difference was found between the two groups at the age of six months and none during the further follow-up of two years. Therefore, a preventive hemifundoplication in CDH patients in general could not be recommended [18]. It seems noteworthy that, in this study, no differentiation between defect sizes was made, as the internationally accepted classification system of CDH defect size was only incorporated after this study was performed [3]. This seems particularly interesting when looking at another study published in 2020, in which 126 CDH patients were retrospectively analyzed, and “severe defect grade” was identified as an independent risk factor for the need for anti-reflux surgery during follow-up. Furthermore, 33% of patients underwent a Nissen fundoplication at a median age of 61 days after CDH repair [19]. “Severe defect grade” was defined as defect size C (>50% of the hemidiaphragm missing) or D (near complete absence of the hemidiaphragm) and liver herniation [3]. To date, the question whether a simultaneous anti-reflux procedure at the time of CDH repair should be performed cannot be answered with certainty since results of different studies are contradictory. Gastroesophageal reflux is not only a problem in the CDH infant but can often occur in adult CDH patients as well [20]. The aim of this study was to assess long-term symptoms of GER and the need for secondary anti-reflux surgery in patients with and without preventive hemifundoplication at initial CDH repair within a standardized follow-up program, to identify risk factors, and to suggest standard examinations for those patients particularly vulnerable to long-term problems associated with their primary disease.

## 2. Materials and Methods

For this study, 79 patients who formerly participated in a randomized controlled trial with informed consent were eligible. The case group received an additional hemifundoplication and fundopexy at the time of surgical repair of the diaphragmatic defect, whereas the control group received only the surgical reconstruction of CDH [18]. Due to the absence of the CDH classification system at the time of initial surgery, referring to surgical reports, all diaphragmatic defects were retrospectively categorized into defect size A to D in accordance with the CDH study group [3]. Defect sizes A and B were then summarized as small defects and defect sizes C and D as large defects. After discharge, CDH patients were seen regularly in our prospective follow-up program at defined intervals. At each visit, a structured case history was raised and symptoms for gastroesophageal reflux were carefully assessed. Symptoms of reflux were categorized into two groups: the first was (1) none to mild symptoms (i.e., regurgitation, occasional vomiting, temporary use of proton pump inhibitors (PPI)), which had a relatively low impact on patients’ quality of life, and the second category, which was defined as (2) moderate or severe symptoms (recurrent vomiting, recurrent pulmonary infections or obstructive lung function disorders, failure to thrive, long-term use of PPI, pathologic results in 24 h (impedance) pH metry, or necessity of anti-reflux surgery), which had a relevant impact on patients’ quality of life. Secondary anti-reflux procedures included a hiatoplasty and hemifundoplication (Thal’s procedure) in the majority of patients and, in some infants only the insertion of a jejunal feeding tube. Three time intervals were defined in order to demonstrate symptoms in infants and toddlers (0–2 years), in kindergarten children (>2–6 years), and in school children (> 6 years of age). Body weight was assessed at each visit, and percentiles and z-scores were calculated [21]. Statistical analysis was performed using GraphPad Prism version 8.4.3 for Windows (GraphPad Software, San Diego, CA, USA, www.graphpad.com (accessed on 10 March 2022)). Continuous variables were reported as median or mean and compared between the two groups using 2-sample independent *t*-test or Mann–Whitney *u*-test (non-normal data). Contingency tables were analyzed with Fisher’s exact test or chi-square test where applicable. *p*-values < 0.05 were considered significant.

Predictors for the need of anti-reflux surgery during follow-up were identified using logistic regression analysis, for which the following variables were determined: defect size, patch repair of CDH, intrathoracic liver or stomach herniation, abdominal wall patch, and ECMO. All variables were included in the univariable und multivariable model.

This study was approved by our local ethic committee (2018-592N-MA), and informed consent was obtained from all parents.

## 3. Results

### 3.1. Patient and Surgical Characteristics

Of the 79 patients enrolled in the original study, all had left-sided CDH and were operated by median laparotomy. Thirteen patients (16.5%) had to be excluded from long-term analysis: nine children died within the first year of life (11.4%), and two (2.5%) did not participate in our follow-up program. In two others (2.5%), a late diagnosis of Cornelia-di-Lange syndrome made the assessment of gastroesophageal symptoms unreliable, as these patients suffer from feeding difficulties due to neurologic impairment; thus, they were also excluded [18]. Overall, 66 patients (30 male, 36 female) were analyzed with a follow-up time of a median of ten years (range 0.6–16 years), of whom 31 patients (47%) received a preventive anti-reflux surgery at the time of diaphragmatic repair. Patient and surgical characteristics are displayed in Table 1. CDH repair was performed after stabilization of the neonate and ECMO decannulation. Duration of ECMO was median eight days (range 6–14 days). In all patients with large defect size, the diaphragm was reconstructed using a cone-shaped polytetrafluoroethylene patch to avoid closure under tension, as described by our group in 2005 [22]. About 70% of patients were classified as having large defect size. Nevertheless, the percentage of patients receiving preventive anti-reflux surgery was equally distributed in patients with small and large defects.

### 3.2. GER Symptoms at Different Time Intervals

During the first two years of life, 26% of patients presented with symptoms of moderate or severe gastroesophageal reflux. This rate decreased significantly to 7% at the age between two to six years (*p* = 0.007) and remained stable in 6% of children older than six years. Characteristics of patients with moderate or severe symptoms of GER are summarized in Table 2. A significant difference was detected for defect size of CDH, showing significantly more moderate or severe GER during the first two years of life in patients with large defects. Patients with patch repair of the diaphragmatic defect also had considerably more moderate or severe GER symptoms than patients whose diaphragmatic defect was repaired primarily, but this did not quite reach significance (*p* = 0.051). None of the patients with small defects and 8% of patients with large defects showed symptoms in older age.

There was no statistically significant difference concerning moderate or severe symptoms in patients that had received a primary fundoplication at the time of diaphragmatic repair at any age. Those patients with large defects and a preventive fundoplication were analyzed to see whether this suspected high-risk group benefited from the preventive fundoplication, but no significant difference could be detected. Intrathoracic liver and stomach herniation and ECMO therapy did not affect GER at any time point.

### 3.3. Surgical Characteristics of Patients with and without Secondary Antireflux Surgery

Twelve patients (18%) required an anti-reflux surgery or feeding tube insertion during follow-up at a median age of 9.5 months (range 3 months–16 years) either due to severe feeding problems, failure to thrive or pulmonary symptoms. Four patients received a hiatoplasty and hemifundoplication at the ages of 9, 10, 26, and 30 months, respectively. Two patients received a fundoplication in combination with the establishment of a jejunostomy and one in combination with a gastrostomy at the ages of 5, 7, and 15 months, respectively. In one patient, at the age of five months, a hiatal hernia was closed. Two patients solely received a jejunostomy within the first nine months of life, of which one suffered from a CDH recurrence at the age of two years. At surgical repair of CDH recurrence, the jejunostomy was converted to a gastrostomy. Two patients received a hiatoplasty and hemifundoplication at school age (10 and 16 years); both had developed a secondary hiatal hernia after initial large defect size and patch implantation. Characteristics of patients with and without secondary anti-reflux surgery during follow-up are summarized in Table 3.

The variables large defect size, CDH repair with patch, intrathoracic liver, and stomach herniation were significantly associated with secondary anti-reflux surgery, as three of these variables were present in all patients who required a secondary intervention due to GER. In logistic regression analysis, intrathoracic liver herniation remained as the only independent risk factor for this (OR = 6.3, 95% CI 1.3–31.3, *p* = 0.015). In patients who required secondary anti-reflux surgery, a higher rate of primary fundoplication was noted, but this difference did not quite reach statistical significance.

### 3.4. Weight Development of Patients at Different Time Intervals

The mean weight percentile of all patients in our study cohort was 14.2% for the first 24 months of life, 20.2% between 24–72 months, and 22.2% older than 72 months, respectively. When dividing the cohort into patients with small and large defects, a highly significant difference could be detected revealing a markedly lower body weight for children with large CDH in all age groups. Furthermore, no significant effect of preventive anti-reflux surgery on weight gain could be encountered neither in the whole cohort nor in the subgroup of patients with large defects. In patients who had received a preventive anti-reflux surgery, mean body weight was constantly lower than that of patients who had received closure of the diaphragmatic defect alone. These data are visualized in Figure 1. Regarding the effect of secondary anti-reflux surgery, which in most cases in our cohort was performed during the first two years of follow-up, data show that initial significant weight differences approximate with age, and no significant difference at the age older than 72 months can be detected. All data are displayed in Table 4.

## 4. Discussion

Long-term follow-up of patients with and without preventive anti-reflux procedure at initial CDH repair revealed that patients with small defects did not present with any significant gastroesophageal reflux symptoms beyond the age of two years. Patients with large defects had a prevalence of 35% of moderate or severe GER symptoms during the first two years and significantly less thereafter. Secondary anti-reflux procedures were only necessary in patients with large defects and significantly more often in patients with intrathoracic liver and stomach herniation as well as patch repair of the diaphragmatic defect. Liver herniation was the only independent risk factor identified in multivariate regression analysis. Primary fundopexy and hemifundoplication did not reveal a protective effect regarding the occurrence of GER symptoms, the need for secondary anti-reflux surgery or the gain of body weight neither in the short nor in the long term.

In our cohort, the prevalence of clinically relevant symptoms of GER of 26% within the first two years, 7% between two and six years, and 6% thereafter seems relatively low when compared to other studies. Koivusalo et al. conducted a follow-up of 26 CDH patients and, by assessing clinical symptoms AND performing endoscopies and 24 h pH metries, detected a prevalence of significant gastroesophageal symptoms of 27% (7/26) at six months, 42% (11/26) at one year, 53% (8/15) at three and five years, and 55% (5/9) at ten years of age, respectively [23]. Similar results were found by Arena et al., reporting GER by 24 h pH monitoring in six of eleven CDH patients at the age of 4.5 years and in five of fifteen patients at the age of 21 years [24]. In the only review and meta-analysis on GER after CDH repair, involving a total of 1051 infants and 389 children older than one year, a pooled prevalence of gastroesophageal reflux disease was estimated at 46.4%—with 52.7% in children under the age of one year and 35.1% in older children, respectively. In general, a higher prevalence of GER was detected after evaluation with impedance and pH metry [13]. In a patient-led survey by CDH UK, a registered charity for CDH patients governed by a volunteer committee, 62–92% of 151 participants reported GER symptoms [25]. Interestingly, “three fourths of the responders did not agree with the statement that feeding problems improved with time”.

In contrast, others reported decreasing symptoms with ongoing age, which is also the natural course of GER in healthy children [26,27]. We also observed a decreasing prevalence of GER symptoms with age in our CDH cohort with a median follow-up of ten years. CDH patients with large defects had a significantly higher incidence of gastroesophageal reflux symptoms especially during the first two years of life, whereas none of the patients with small defects displayed severe gastroesophageal reflux beyond two years of age. In contrast, GER was observed significantly more often after primary repair by Kamiyama et al. [28]. Other authors also identified primary CDH repair as being associated with a high prevalence of severe GER and identified CDH repair with tension as a risk factor for severe GER [24,29,30]. A possible explanation could be the difference in the type of CDH repair between these studies. Koivusalo et al. reported a patch-rate of 38% compared to 76% in our cohort [23]. We followed the concept of a tension-free reconstruction of the diaphragm and thus used patches liberally also in B-defects. Koivusalo chose a subcostal laparotomy and did not specify the kind of diaphragmatic patch that was implanted, whereas a cone-shaped patch as previously described was implanted via a median laparotomy in our cohort [22]. The main advantage of the cone-shaped patch is the increase of the volume of the hypoplastic abdominal cavity inherent to CDH. This allows an anatomical repositioning of the abdominal organs with an at least near-normal angle of His. Furthermore, intraabdominal pressure after repositioning of the abdominal viscera is reduced, which has been suspected as one of the many causes for a higher prevalence of GER after CDH repair. Implantation of plane patches or primary reconstruction of the diaphragm with tension results in a steep diaphragm and unanatomical repositioning especially of the spleen and stomach, resulting in a flattened angle of His (Figure 2). All of this could promote the development of GER symptoms and feeding difficulties in the short and persistence of problems in a higher degree in the long term. In 1995 Kieffer et al. already hypothesized different anatomical and surgical factors contributing to the development of GER in CDH neonates: prenatal disturbances of esophageal motility due to compression by viscera herniation, altered anatomy of the gastroesophageal junction, and loss of the angle of His especially in patients with intrathoracic stomach herniation, strain on the diaphragmatic crura in cases with diaphragmatic reconstruction under tension and a high gradient between positive intraabdominal pressure and negative intrathoracic pressure after repositioning of the abdominal viscera [31]. The type of CDH repair may thus contribute to the risk of developing GER. Over time, the cone-shaped patch flattens with growth and hereby reduces tension on the diaphragmatic crura. This might be the reason for the reduced long-term rate of severe GER symptoms in our CDH cohort despite a patch-rate of 100% in large defects. In contrast to our findings, severe CDH has been reported to be associated with the need for secondary anti-reflux surgery in 64% in other cohorts [19].

Chamond et al. reported on a beneficial effect of preventive anti-reflux procedures at the time of initial CDH repair in patients with liver herniation and patch repair of the diaphragmatic defect at a follow-up time of one year. In their cohort of 36 CDH patients, GER symptoms were significantly reduced in patients with large defects who received preventive anti-reflux surgery (17.6% vs. 52.6%, *p* = 0.04) [17]. In contrast to this report, a recent prospective multi-institutional cohort study of 726 patients conducted by Montalva et al. concluded that a preventive fundoplication at the time of surgical repair of CDH was not avoiding secondary anti-reflux surgery during follow-up [32]. Similar to their findings, in our cohort, the group of CDH patients with large defects also did not benefit from preventive fundoplication at CDH repair, as gastroesophageal reflux symptoms tended to present even more often in this group during the first two years of follow-up. Furthermore, prophylactic fundoplication did not prevent the need for another anti-reflux surgery during follow-up. In patients with preventive fundoplication, the rate of secondary anti-reflux surgery during follow-up was even higher although it did not reach statistical significance. One possible explanation could be that in neonates with large, left-sided diaphragmatic defects, the left-sided crus of the diaphragm is also hypoplastic, and a primary hiatoplasty is often not possible due to the lack of diaphragm. Thus, any other “preventive” anti-reflux procedure at the time of diaphragmatic reconstruction may prove to be insufficient in patients with large defect sizes in the long term. Furthermore, natural growth or diaphragmatic reconstruction under tension can cause distraction of the hypoplastic crus from the esophagus and eventually lead to secondary hiatal hernia. We hypothesize that a primary anti-reflux procedure may even promote stomach herniation in patients with large defects developing secondary hiatal hernia, which could not only explain more severe symptoms of GER but also the reduced gain of weight and higher rate of secondary anti-reflux surgery in this patient group.

The reported necessity for secondary anti-reflux surgery after CDH repair varies greatly concerning percentage and time in mainly retrospective studies. Diamond et al. reported that 15% of 86 patients required anti-reflux surgery during a follow-up time of 55.1 months with a mean time from CDH repair to GER intervention of 4.7 months, whereas 33% of 126 patients required a Nissen fundoplication at a median age of 61 days, as reported by Guglielmetti et al. [15,19]. Thus, a substantial subset of CDH patients seems to undergo secondary anti-reflux procedures rather early during the first year of life. In our cohort, there was a need for secondary anti-reflux surgery in only 18% of patients during a much longer follow-up time. In our experience, solely CDH patients with large defects required secondary interventions and intrathoracic liver herniation was identified as the only independent risk factor. This is in accordance with the findings of Verbelen and Diamond, who also identified liver herniation as an independent risk factor [12,13]. Others have identified intrathoracic stomach herniation as a risk factor [16,26,30,31,33]. In our cohort, secondary anti-reflux procedures were related mostly to feeding difficulties in the first year of life. Two of these patients (17%) developed mild GER symptoms at the age of 10 years and could be managed conservatively. So far, no patient in our cohort was subject to a re-fundoplication due to GER recurrence. The reported re-fundoplication rate in infants with GER as a comorbidity of congenital anomalies (esophageal atresia, congenital diaphragmatic hernia, and others), neurological disorders, or primary GER in literature can be as high as 24% [34].

GER seems to be a relevant problem also beyond childhood. A prevalence of late GER of about 16% was reported in adolescents [24,33]. Koivusalo et al. assessed the health-related quality of life in 69 adults after CDH repair using a validated questionnaire. The median age of participants was 39 years, and the prevalence of gastroesophageal reflux symptoms was 20%, which is significantly higher than in the control group of healthy adults (2%) [35]. Even though the prevalence is higher than in our cohort, severity of diaphragmatic defects must have been less, as of the 69 patients assessed, 24 defects were classified as diaphragmatic eventrations, and only one of the 45 diaphragmatic defects required a patch repair. It has to be critically evaluated whether primary closure of a diaphragmatic defect with tension may have promoted the development of GER in the long term in this cohort. Still, this study suggests an increase in reflux symptoms with age in long-term survivors of CDH. In accordance, Vanamo et al. found that 63% of 60 adult CDH patients at a mean age of 29 years reported gastroesophageal reflux symptoms, and 54% of 41 endoscopically assessed patients presented with esophagitis or even Barrett’s esophagus [20]. This gains even more importance when considering the study by Caruso et al., who were able to show a high prevalence of gastroesophageal reflux with esophagitis performing multichannel intraluminal impedance and pH metries in 36 CDH patients at the age of six months and five years. Remarkably, reflux was detected in 72% and 45% of asymptomatic patients, respectively, indicating that CDH patients can suffer from silent and mainly nonacidic reflux [30]. These findings were confirmed by di Pace et al., who also detected a pathological GER in 86% of 30 patients at a median age of 5.2 years, while 46% were asymptomatic. GER was nonacidic, postprandial, short-term, and only reaching the distal esophagus in the majority of cases [29]. Morandi et al. evaluated twelve adolescents: only three had a pathological questionnaire, while nine showed esophagitis ≥ grade 2, and one of these was Barrett’s esophagus on endoscopic evaluation. In contrast, histology revealed severe esophagitis in only two patients [27]. Thus, there seems to be a gap between findings on diagnostic evaluation and self-reported GER symptoms or those picked up by validated questionnaires.

It is known that CDH patients generally present with a low body weight during childhood, which can be caused by several circumstances [36,37]: these children have a higher energy consumption due to strained breathing efforts as they are compensating lung hypoplasia [38]. Secondly, adequate oral food intake requires substantial and continuous effort by the parents, as CDH patients often present as “picky eaters”. Lastly, GER can be a major cause for failure to thrive, which can affect neurocognitive development negatively [11]. GER, therefore, has a substantial effect on the quality of life of CDH survivors. In a cohort with primary CDH repair, Arena et al. observed a significantly lower weight in patients with GER at one year of age but no growth impairment in the long term [24]. Chamond et al. reported growth retardation in 12% [17], whereas Dariel did not detect growth disorders after preventive anti-reflux surgery [39]. However, Montalva et al. could associate primary fundoplication with higher rates of failure to thrive at discharge (81% vs. 51%, *p* = 0.03) as well as at the age of six months (81% vs. 45%, *p* = 0.008) and showed higher rates of dependency on tube feeding in these patients up to the age of two years (65% vs. 26%, *p* = 0.004) [32]. Correspondingly, our data show that preventive anti-reflux surgery has no positive effect on weight gain in these patients, even in patients with large defects of the diaphragm. Quite contrarily to the anticipated effect, children with preventive fundoplication presented with markedly lower body weights, whereas secondary anti-reflux surgery during follow-up had a positive effect on patients’ body weight. As secondary fundoplication was performed in a target-oriented manner in patients actually suffering from severe GER, initially significantly lower body weight in these patients approximated with the body weight of patients that did not require secondary anti-reflux surgery in the long term. This supports the claim for a tailored approach for anti-reflux surgery in CDH patients, especially focusing on patients with large defects and performing surgery after severe GER has been diagnosed.

Our results are based on history taking and assessment by upper GI study mainly in younger patients, endoscopy, and/or 24 h (impedance) pH metry in older patients with suspected GER. The etiology of GER in younger and older children might be different although large defect size is the common underlying condition. Large defect size is a surrogate parameter for CDH severity, namely lung hypoplasia with longer ventilation time, prolonged stay in the intensive care unit and in the hospital, and thus longer administration of sedatives. Time to full enteral nutrition may be prolonged in these children, and analgosedatives are known to reduce intestinal motility and promote gastroesophageal reflux. Furthermore, respiratory distress may lead to aerophagia and promote GER. Additionally, delayed gastric emptying as well as gastric and esophageal dysmotility may play a role [24,29]. All these factors may worsen GER in CDH patients who display some predisposing anatomical factors, as explained in detail above. In younger children, the type of diaphragmatic patch repair might therefore play a substantial role in the findings of different centers, and avoidance of diaphragmatic reconstruction under tension seems to be a key issue. In older children, a secondary hiatal hernia can develop with growth after implantation of a diaphragmatic patch in large defects [1]. Since GER symptoms develop slowly or have been there ever since, they are often recognized neither by the patient nor by the parents, and therefore, as such, bear the risk of mucosal dysplasia and secondary malignancy in the long term [10,16]. In our experience, children with chronic reflux typically adapt and eat small portions throughout the whole day. Typical symptoms of GER may therefore be subtle and may not be picked up validly by questionnaires or anamnesis. In older children, obstructive lung function disorders and slow thriving might be the only hints. This underlines the importance of a structured follow-up program through childhood, adolescence and transition into adulthood with careful history taking and identification of patients at risk. Further diagnostic evaluation should preferably comprise endoscopy and 24 h impedance measurement due to the more sensitive detection of nonacidic reflux episodes and esophageal dysmotility. The threshold for further investigation should be low in patients showing secondary hiatal hernia or failure to thrive on routine follow-up. In completely asymptomatic patients with normal growth and weight gain but with initially large defects requiring patch repair and with intrathoracic liver and stomach herniation, a routine (impedance) pH metry should be considered, at the latest, at 18 years of age to screen for silent reflux with the consecutive risk of Barrett’s esophagus or secondary malignancy. Transitional follow-up programs should be established in all reference centers, and regular follow-up of GER ought to be implemented according to the results of GER screening at the age of 18.

Our study is limited by the monocentric setting and relatively small number of patients. Yet, it is the only prospective randomized controlled trial concerning preventive anti-reflux procedures in CDH patients with a long-term follow-up of a median of ten years. Another limitation is the retrospective classification of defect sizes from surgical reports, as the reporting system was not available at the time of initial study implementation. Nevertheless, a patch-rate of 100% in patients with large defects seems reasonable. Within our study, no routine GER investigation with endoscopy or 24 h (impedance) pH metry was performed in all patients but was limited to those with suspected GER. Therefore, some asymptomatic patients with silent or nonacidic reflux might have been missed so far, and the rate of GER might be underreported.

In conclusion, severe GER necessitating secondary anti-reflux procedures in CDH patients was significantly correlated to large defect size with initial liver and stomach herniation and requiring patch repair in our cohort. Initial tension-free reconstruction of the diaphragm and the use of a cone-shaped patch may be beneficial regarding the short- and long-term prevalence of GER. On the other hand, preventive fundoplication at the time of initial CDH repair did not reduce GER symptoms and the need for secondary anti-reflux surgery or improve the gain of body weight in patients with small or large defect size. A structured follow-up program until adolescence and transition into adulthood with careful appreciation of subtle GER symptoms and routine evaluation of GER, including endoscopy and impedance measurement, should be recommended especially for high-risk patients.

## Figures and Tables

**Figure 1 children-09-01137-f001:**
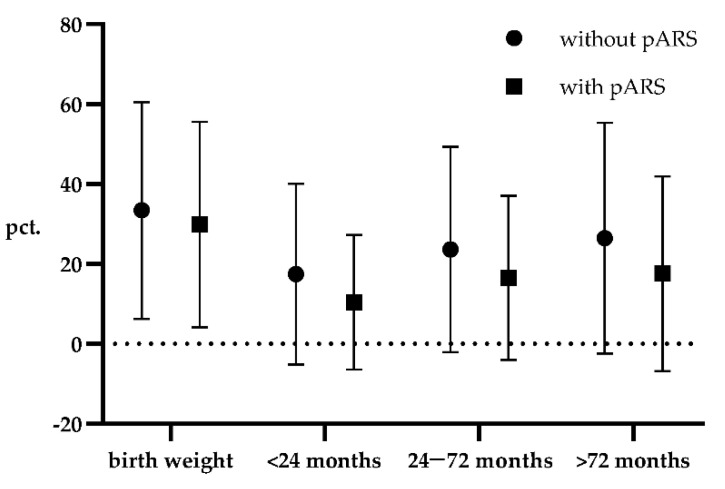
Mean of weight percentiles of patients with and without preventive anti-reflux surgery at different time intervals. Error bars indicate SD. pARS, preventive anti-reflux surgery; pct., percentile.

**Figure 2 children-09-01137-f002:**
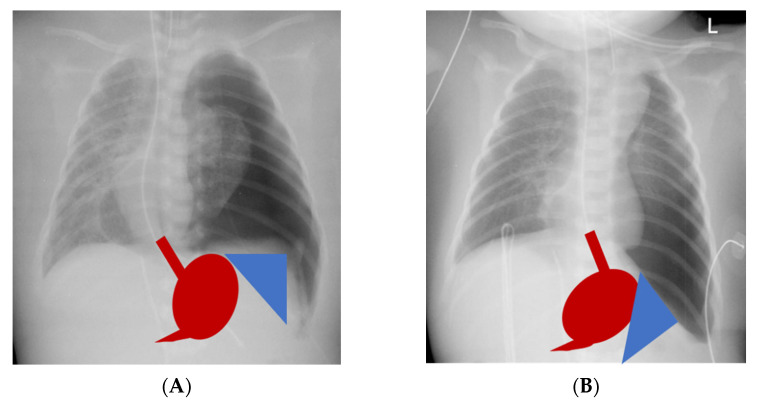
Differences in repositioning of stomach (red) and spleen (blue) after repair of congenital diaphragmatic hernia: (**A**) with a cone-shaped patch, physiologic repositioning, and near-normal angle of His; (**B**) after primary repair under tension or implantation of a plane patch, altered anatomy, and flattened angle of His.

**Table 1 children-09-01137-t001:** Patient and surgical characteristics of the study-cohort.

	Study Cohort (*n* = 66)	Small Defects (*n* = 20)	Large Defects (*n* = 46)	*p*
Prenatal diagnosis	51 (77.3%)	11 (55%)	40 (87%)	**0.009**
Female gender	30 (45.5%)	10 (50%)	20 (43.5%)	0.117
Birth weight (g) SD	3015 536	3177 598	2950 497	0.171
Birth length (cm) SD	50 2.9	51.1 3.2	50 2.7	0.103
Gestational age (weeks) range	37 + 6 32 + 1–41 + 4	38 + 3 35 + 1–41 + 4	37 + 6 32 + 1–41 + 4	0.558
Liver herniation	34 (52%)	2 (10%)	32 (70%)	**0.009**
Stomach herniation	52 (79%)	10 (50%)	42 (91%)	**0.0004**
ECMO therapy	18 (27%)	2 (10%)	16 (35%)	0.069
CDH repair with patch	50 (76%)	4 (20%)	46 (100%)	**<0.0001**
Abdominal wall patch	8 (12%)	0	8 (17.4%)	0.094
Preventive ARS	31 (47%)	8 (40%)	23 (50%)	0.593

Bold: highlight significant differences.

**Table 2 children-09-01137-t002:** Characteristics of patients with moderate or severe symptoms of GER at different time intervals.

	<24 Months (*n* = 66)		24–72 Months (*n* = 58)		>72 Months (*n* = 50)	
	Moderate or Severe GER Symptoms	*p*	Moderate or Severe GER Symptoms	*p*	Moderate or Severe GER Symptoms	*p*
Small defects	1/20 (5%)	**0.013**	0/18 (0%)	0.545	0/13 (0%)	0.558
Large defects	16/46 (35%)		3/40 (8%)		3/37 (8%)	
Without ECMO	13/48 (27%)	0.763	2/43 (5%)	>0.99	1/38 (3%)	0.139
ECMO	4/18 (22%)		1/15 (7%)		2/12 (17%)	
Primary repair	1/17 (6%)	0.051	0/16 (0%)	0.554	0/16 (0%)	0.544
Patch repair	16/49 (32%)		3/42 (7%)		3/36 (8%)	
No liver herniation	5/32 (15%)	0.093	1/29 (3%)	>0.99	0/24 (0%)	0.491
Liver herniation	12/34 (35%)		0/23 (0%)		2/26 (8%)	
No stomach herniation	1/14 (7%)	0.093	0/13 (0%)	>0.99	0/10 (0%)	>0.99
Stomach herniation	16/52 (31%)		3/45 (7%)		3/40 (8%)	
Without pARS	7/35 (20%)	0.276	2/30 (7%)	>0.99	2/25 (8%)	>0.99
With pARS	10/31 (32%)		2/29 (7%)		1/24 (4%)	
- Large defects w/o pARS	6/23 (26%)	0.353	2/20 (10%)	>0.99	2/18 (11%)	>0.99
- Large defects w/pARS	10/23 (43%)		2/21 (10%)		1/18 (6%)	

Bold: highlight significant differences.

**Table 3 children-09-01137-t003:** Surgical characteristics of patients with and without secondary anti-reflux surgery.

	Study Cohort (*n* = 66)	Secondary Anti- Reflux Surgery (*n* = 12)	No Secondary Anti- Reflux Surgery (*n* = 54)	*p*
Large defect size	46 (70%)	12 (100%)	34 (63%)	**0.012**
CDH repair with patch	50 (76%)	12 (100%)	38 (70%)	**0.03**
Liver herniation	34 (52%)	10 (83%)	24 (44%)	**0.015**
Stomach herniation	52 (79%)	12 (100%)	40 (74%)	**0.019**
ECMO therapy	18 (27%)	4 (33%)	14 (26%)	0.722
Abdominal wall patch	8 (12%)	2 (17%)	6 (11%)	0.594
Preventive anti-reflux surgery	31 (47%)	9 (75%)	22 (41%)	0.053

CDH, congenital diaphragmatic hernia; ECMO, extracorporeal membrane oxygenation; Bold: highlight significant differences.

**Table 4 children-09-01137-t004:** Percentiles and z-scores of body weight at different time intervals.

	<24 Months				24–72 Months				>72 Months			
	pct.	*p*	z-Score	*p*	pct.	*p*	z-Score	*p*	pct.	*p*	z-Score	*p*
Small defects	28.5	**<0.0001**	−0.63	**<0.0001**	34.2	**0.002**	−0.68	**0.004**	38.2	**0.004**	−0.51	**0.002**
Large defects	8.0		−1.89		13.9		−1.71		15.5		−1.70	
Without pARS	18	0.161	−1.43	0.521	23.7	0.264	−1.22	0.305	26.5	0.229	−1.15	0.247
With pARS	10.5		−1.61		16.6		−1.57		18		−1.57	
- Large defects without pARS	10.9	0.119	−1.78	0.389	18.9	0.086	−1.37	0.057	19.2	0.286	−1.45	0.216
- Large defects with pARS	5.0		−2.01		9.0		−2.06		11.7		−1.95	
Without secondary ARS	16.7	**0.031**	−1.33	**0.004**	22.6	0.08	−1.25	**0.045**	24.3	0.169	−1.25	0.169
With secondary ARS	2.8		−2.33		7.7		−2.15		10.0		−1.95	

Bold: highlight significant differences.

## Data Availability

The data presented in this study are available on request from the corresponding author.

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
