# Peer review of "Long-Term Evaluation of Gastroesophageal Reflux in Neonates with and without Preventive Anti-reflux Surgery at the Time of Congenital Diaphragmatic Hernia Repair"

_children, 2022, doi:10.3390/children9081137_

Round 1

Reviewer 1 Report

This manuscript details a well designed study of an interesting topic.  While preventive anti-reflux surgery is not routinely done during CDH repair in the US, it is a topic that has been of interest, given the high incidence of GER in patients with CDH.  It was interesting to see demonstrated the risk factors for moderate-severe GER in patients who have had CDH repair in infancy.  It is  an important finding that anti-reflux surgery done on a preventive basis along with CDH repair did not have a significant effect on later need for anti-reflux surgeries, similarly, it was interesting to see that there were techniques in the CDH repair itself that could possibly decrease the risk of moderate-severe GER in the future. 

There are some grammatical issues throughout, especially in the introduction, with very long sentences that would be much easier to read if broken into some shorter ones.

The manuscript, on line 55 makes reference to ALTE, which is a term no longer in use, and has been replaced by BRUE.

Overall, well done study and well written report of such.

Reviewer 2 Report

The authors reported long-term outcome subsequent to RCT which compared 2 groups with or without preventive antireflux surgery at the time of congenital diaphrgmatic hernia repair. The theme is important and the draft was well written. 

Reviewer 3 Report

Authors performed meticulous analyses and discussion about GER in CDH patients with or without preventive antireflux surgery at the time of CDH repair.

I would like to ask authors one thing. You and some previous reports showed that preventive fundoplication at the tine of CDH repair did not reduce GER and the need for secondary antireflux surgery. What kind of causes can you think of this? Are there any anatomical problems of CDH neonates, or surgical factors of simultaneous procedure of CDH repair? I recommend you would discuss about it in the discussion part.

Reviewer 4 Report

A well-written manuscript, with clear patient enrollment, follow-up and comparison of two groups.

Reviewer 5 Report

In this study, the authors conducted a randomized controlled trial to see the long-term outcomes between the case group (with a preventive hemifundoplication at the time of CDH repair) and the control group (without a preemptive antireflux procedure). However, they don't show any comparison between those groups. Besides, their "preventive" antireflux procedure was not "preventive" since 32% of patients with this procedure still had GER symptoms, which is a greater rate than patients without this procedure. Regarding this context, this "RCT" does tell only their "preventive" procedure was not "preventive." So that their conclusion that "preventive fundoplication had no benefit with small nor with large defects" was fallacious. The authors should re-design this study. 

Round 2

Reviewer 5 Report

I am afraid that the authors didn't address my comments in this revised manuscript.

They used data that they collected in the previous RCT. However, they randomized the cohorts in terms of with or without preventive ARS, not small or large defects in their initial RCT study. If they argued their study was RCT for long-term follow-up,  they should have presented the data comparing with or without preventive ARS, not small or large defects shown in Table 1. 

The authors argued in the reply that the "preventable" ARS was preventable in the first six months of age, although this study showed it had no impact on GER symptoms already in the first two years. I am wondering why it happened. What was the rate of recurrence of GER after ARS in their institution? 

They compared the cohorts with or without the need for "secondary" ARS in Table 3. What was the "secondary antireflux surgery" for patients without primary "preventive" surgery? Interpretation of this comparison 

was confusing. Would "Preventive ARS"  affect the requirement of secondary surgery?

 Only the result shown in Figure 1 was the outcome of this RCT. Otherwise, they conducted a retrospective analysis using the data collected in the previous RCT.

 I assume they want to show severe CDH (large defect, liver and stomach herniation) may be associated with severe GER both in the short term and long term regardless of additional "preemptive ARS." If they want to show it as the outcome of RCT, they should show the comparison between the cohorts with or without preemptive ARS, and they need to focus on why the preemptive ARS would fail (recurrence of GER) in the long run.

If they want to focus on GER in the long-term using retrospective defect size data, it is a retrospective analysis using the same cohort as their previous RCT.